# The Relationship Between Contrast Associated Nephropathy and Coronary Collateral Circulation in very Old Patients

**DOI:** 10.3390/medicina56030099

**Published:** 2020-02-27

**Authors:** Tarik Yildirim, Birol Ozkan, Gokhan Alici, Seda Elcim Yildirim, Onursal Bugra, Hasan Kadi

**Affiliations:** 1Cardiology Department, Balikesir University Faculty of Medicine, 10100 Balikesir, Turkey; sedaelcimdurusoy@gmail.com (S.E.Y.); drhkadi@gmail.com (H.K.); 2Department of Cardiology, University of Health Sciences Turkey, Kartal Kosuyolu High Specialty Educational and Research Hospital, 34865, Istanbul, Turkey; drbirolozkan@hotmail.com (B.O.); gokhanalici@yahoo.com (G.A.); 3Department of Cardiovascular Surgery, Balikesir University Faculty of Medicine, 10100 Balikesir, Turkey; onursalbugra@gmail.com

**Keywords:** Contrast associated nephropathy, Coronary collateral circulation, Endothelial function, Very old patient

## Abstract

*Background:* The aim of this study was to investigate whether there is a relationship between coronary collateral circulation (CCC) and contrast associated nephropathy (CAN) in very elderly patients. *Methods:* Patients aged 90 years or older with at least one major occlusion of the coronary artery proximal or mid-section were included in the study. CCC was graded according to the Rentrop classification. CAN was defined as an increase in blood creatinine value of 25% or more on the second day after coronary angiography. *Results:* Thirty-six patients who met the study criteria were included in the study. In the study group, CAN developed in 12 patients (CAN (+) group), 24 patients did not develop CAN (CAN (−) group). The creatinine levels before coronary angiography were 1.05 ± 0.12 in the CAN (−) group and 1.22 ± 0.14 in the CAN (+) group. Baseline creatinine values were significantly higher in the CAN (+) group (*p* = 0.001). The contrast agent used in the CAN (+) group was significantly higher (*p* = 0.001). In the CAN (+) group, nine patients (43%) had poor collateral circulation, whereas only three patients (20%) had well-developed collateral circulation. In a logistic regression analysis, the collateral class was not a risk factor for CAN, whereas contrast agent volume and basal creatinine were independent predictors of CAN. *Conclusion:* We found that CCC grade was not associated with the development of CAN in very old patients, but the amount of contrast agent and pre-procedure creatinine values were independent variables in the development of CAN.

## 1. Introduction

There are many collateral vessels connecting the major coronary arteries in the human heart [1]. Coronary collateral circulation (CCC) is an alternative way of delivering blood to the ischemic region in case of occlusion of the coronary artery. As a result of advances in molecular biology and genetic science, some clues have been obtained regarding the mechanisms of collateral development, but the exact mechanism has not been fully elucidated. The role of the endothelium in the development of congenital collateral vessels into mature arteries has been emphasized, and in both experimental and clinical studies, it has been shown that intact and functional endothelium has a very important role [2,3,4]. In this context, a new epithelium-derived pigment factor (PEDF) has been revealed as having a key role, as well as well-known factors such as nitric oxide (NO) and vascular endothelial growth factor (VEGF) [5,6]. In addition, vascular endothelial cadherin (VE-cadherin), the strongest cadherin, is expressed by endothelial cells [7]. Contrast associated nephropathy (CAN) is an entity that occurs after contrast exposure is an acute renal failure that is usually reversible. CAN has been reported to prolong the length of hospital stay, leading to an increase in mortality and morbidity [8]. Although CAN is seen in 5–12% of those exposed to contrast media, this rate increases by up to 70% in high-risk patients [9,10,11,12]. Although its pathophysiology has not been fully elucidated, the direct toxic effect of contrast agent on renal tubular cells, renal medullar hypoxia, impaired vasoconstriction/vasodilatation balance, and underlying endothelial dysfunction are the main mechanisms responsible for the development of CAN.

The aim of this study was to investigate whether there is a relationship between coronary collateral circulation and contrast associated nephropathy in very elderly patients.

## 2. Materials and Methods

### 2.1. Patients

This study started after the approval of the local ethics committee. Patients who underwent coronary angiography in the coronary angiography laboratories of Balikesir University Faculty of Medicine Cardiology Department and the Cardiology Department of Balikesir State Hospital between January 2012 and May 2019, were retrospectively analyzed. Patient information was obtained from electronic records. The study protocol was approved by the Balikesir University Ethics Committee (Approval number: 2016/73, Date:04/05/2016).

### 2.2. Inclusion Criteria

(1) Patients who underwent diagnostic coronary angiography; (2) patients with complete occlusion of at least one major coronary artery proximal or mid-section on coronary angiography; (3) patients whose blood creatinine levels were measured before coronary angiography, and on the 2nd day after coronary angiography, were included in the study.

### 2.3. Exclusion Criteria

(1) Patients with acute coronary syndromes and patients undergoing primary percutaneous intervention, (2) patients who had undergone percutaneous coronary intervention or underwent coronary artery surgery, (3) patients with functional capacity New York Heart Association (NYHA) class III and IV, (4) patients with moderate-severe valve disease, (5) patients with a blood creatinine level ≥1.5 mg/dL prior to coronary angiography, (6) patients treated to prevent nephropathy triggered by contrast agent before and after coronary angiography (saline infusion, bicarbonate treatment), (7) patients with diabetes and severe hypertension, (8) patients with acute severe renal failure, and (9) patients with previous myocardial infarction were excluded from the study. The study protocol was approved by the local ethics committee.

### 2.4. Coronary Angiography and Grading of Collateral Flow

All patients underwent coronary angiography using the Judkins technique. Significant stenosis was defined as a 50% greater diameter narrowing of any major coronary artery. The collateral flow was graded according to the Rentrop classification [13]. Rentrop 0: No collateral flow, Rentrop 1: Clogged main coronary artery without seeing the side branches, Rentrop 2: Partial view of the occluded main coronary artery, Rentrop 3: Complete view of the occluded main coronary artery. In the presence of multiple occluded vessels, the highest degree of collateral was selected for analysis. If there was more than one collateral flow to the same occluded vessel, the highest grade was used for analysis. In accordance with previous studies, those classified as Rentrop 0 and 1, collateral circulation was deemed insufficient, those classified as Rentrop 2 and 3, coronary collateral circulation was evaluated as sufficient [14]. All coronary angiographies were analyzed by two experienced interventional cardiologists.

### 2.5. Diagnosis of Contrast Associated Nephropathy

Contrast associated nephropathy was defined as an increase in serum creatinine value of 25% or more compared to the initial value within 48 h of exposure to the contrast agent [15].

### 2.6. Statistical Analysis

SPSS for Windows 22 (SPSS Inc. Chicago, IL, USA) statistical package program was used for statistical analysis. The suitability of the data for normal distribution was analyzed by a Kolmogorov–Smirnov test. Categorical variables were given as number and percentages; continuous variables are shown as mean ± standard deviation (mean ± SD). Mann–Whitney-U test was used for the analysis of continuous variables that did not fit the normal distribution. Analysis of continuous variables matching normal distribution was performed by two independent samples *t*-test. Chi-square tests were used to compare categorical variables between groups. Fisher exact test was performed where necessary. The study group was divided into two groups: the CAN (+) group and CAN (−) group. Logistic regression analyses were performed to determine the independent factors affecting the development of contrast associated nephropathy. A *p* value less than 0.05 was considered statistically significant. 

## 3. Results

Between January 2012 and March 2019, 11,280 coronary angiograms were examined in both centers. Thirty-six patients who met the study criteria were included in the study. In the study group, CAN developed in 12 patients (32%) (CAN (+) group), and did not develop in 24 patients (CAN (−) group). fifty percent of the patients in the CAN (−) group and 42% of the patients in the CAN (+) group were female (*p* = 0.642). The mean age (years) was 91.3 ± 1 in the CAN (+) group and 91.2 ± 1 in the CAN (−) group. There was no difference between the two groups in terms of age (*p* = 0.645). Hypertension was present in 50% of the CAN (+) group and 42% of the CAN (−) group (*p* = 0.642). The two groups were similar in terms of using the cardiovascular drug. The number of diseased vessels was similar in both groups (*p* = 0.609). There was no significant difference between the groups in terms of mean hemoglobin value. The creatinine values before coronary angiography were 1.05 ± 0.12 in the CAN (−) group and 1.22 ± 0.14 in the CAN (+) group. Baseline creatinine value was significantly higher in the CAN (+) group (*p* = 0.001). The amount of contrast agent used in the CAN (+) group was significantly higher (*p* = 0.001). Fifteen patients in the study group had well-developed CCC, and 21 had poorly developed CCC. In the CAN (+) group, nine patients (75%) had poor CCC, whereas only three patients (25%) had well-developed CCC. In the CAN (−) group, 12 patients (50%) had poorly developed CCC, while 12 patients (50%) had well developed CCC. There was no significant difference between the groups in terms of collateral development (*p* = 0.282). The baseline characteristics of patients in both groups are shown in Table 1 and the comparison with CCC in Table 2. In the logistic regression analysis performed by accepting CAN as a dependent variable and the volume of contrast agent, body mass index, hypertension, coronary collateral development and basal creatinine level as independent variables, collateral class was not a risk factor for CAN whereas contrast agent volume and basal creatinine value were independent predictors of CAN (Table 3).

## 4. Discussion

The present study is the first to investigate the relationship between CAN and CCC in patients aged 90 years and older with chronic stable coronary artery disease. The main findings of our study; CCC in patients 90 years and older is not associated with CAN. Contrast-induced nephropathy was defined as an absolute increase in serum creatinine value of 0.5 mg/dL or a relative increase of 25% in 48 h after contrast agent exposure compared to pre-contrast value in serum creatinine [16,17]. More recently, it has been suggested that the increase in post-contrast creatinine value of 25% or more compared to the value of pre-contrast creatinine is more useful in defining CAN and that relative increase should be used instead of absolute increase [18]. In our current study, the method proposed by “The Contrast-Induced Nephropathy Consensus Panel” was used for the definition of CAN. In patients exposed to contrast agent, creatinine value reaches its highest value within 2–5 days after contrast agent exposure and returns to pre-procedure value within 10–21 days [19]. Although different time periods after contrast exposure were used for the definition of CAN, serum creatinine value was used in 48 h following contrast agent exposure in most clinical studies [20,21,22]. In our present study, creatinine values obtained from blood samples taken on the 2nd day were used to confirm the diagnosis of CAN. In our study, the incidence of CAN development was found to be 32% and it was quite high compared with the findings of previous studies [9]. This can be explained by the fact that our study group consisted of very old patients. We can explain this result with nephrosclerosis. Indeed, biopsy studies have clearly demonstrated that the number of nephrons decreases with age [23]. 

In a previous study, we showed that there was a close relationship between CAN and CCC in patients between the ages of 18–75 and coronary collateral development was an independent predictor for CAN [24]. However, in this study, we included patients aged 90 and over; In both binary comparisons and logistic regression analyzes, we found that there was no relationship between these two entities. The possible causes and explanations of this finding may be.

### 4.1. Nephrosclerosis, Decreased Nephron Density and Direct Nephrotoxic Effect of Contrast Agent

It has long been a well-known fact that there is a close relationship between nephrosclerosis and hypertension. However, it has been shown that this condition is observed among healthy people who do not have hypertension or hypertension-related diseases and nephrosclerosis is observed more frequently with age in healthy people [25]. At the same time, renal biopsy studies have shown that nephron density decreases with age [23]. Considering that our patient group consisted of high-aged patients, it would be a rational assumption that nephrosclerosis is advanced and nephron number decreases in these patients. We can argue that the use of contrast media on this basis will cause CAN more frequently. In fact, the results of our study showed that more contrast agents were used in the CAN developing group and that the amount of contrast agent used was an independent predictor for the development of CAN. In conclusion, it is a reasonable assumption to suggest that the direct toxic effect of the contrast agent is predominant factor in the development of CAN in very old patients. This hypothesis is supported by the study by Henrich et al. showing that the contrast agent has a direct cytotoxic effect on the kidney [26]. There are also other studies showing the direct toxic effect of contrast agent.

### 4.2. Limitation of the Protective Effect of Endothelial Functions

In the present study, there was no difference between the two groups in terms of CCC in the comparison of CCC groups with and without CAN. Moreover, CCC was not a predictor for CAN development. We can explain these results with decreased endothelial function in very old patients. In our study group, only 1 patient had Rentrop grade 3 collateral circulation. This finding may be a sign that endothelial functions are weakened in patients aged 90 and older. Indeed, studies have shown that collateral flow weakens with age and is associated with a decrease in endothelial function [27,28,29]. Decreased endothelial function may be one of the underlying mechanisms of weakening of the protective role of endothelium in the development of CAN, while on the other hand it is a cause of weakening of the collateral development process.

### 4.3. Limitations of the study

The major limitation of our study is the small number of patients. Although we reviewed 11280 coronary angiography, the reason for the low number of patients was that patients selection criteria were very specific in our study. In addition, the lack of proven nephrosclerosis by renal biopsy and lacking of evaluation of endothelial functions are another limitations.

## 5. Conclusions

We found that the grade of CCC in very elderly patients was not associated with the development of CAN, but the amount of contrast agent and pre-procedure creatinine value were independent variables in the development of CAN in this patient group. As a result, we concluded that advanced nephrosclerosis and decreased nephron count may be a predominant factor and endothelial functions may be less effective in development of CAN in very old patients undergoing coronary angiography. We believe that it is beneficial to evaluate the coronary diagnostic procedures and interventions on the benefit-harm axis in very elderly patients as a result of increased life expectancy through further studies.

## Figures and Tables

**Table 1 medicina-56-00099-t001:** Baseline characteristics of the groups.

Variable/Group	CAN (−) *n* = 24	CAN (+) *n* = 12	*p*
Age, year (mean ± SD)	91.3 ± 1	91.2 ± 1	0.649
Female, *n* (%)	12 (50)	5 (42)	0.642
Smoke, *n* (%)	3 (12.5)	3 (25)	0.350
Hypertension, *n* (%)	12 (50)	5 (42)	0.642
Body mass index, kg/m^2^ (mean ± SD)	24.7 ± 1.9	25.2 ± 1.3	0.340
Asetilsalisilic ascites, *n* (%)	39 (81)	47 (80)	0.838
ACE-ARB, *n* (%)	5 (21)	1 (8)	0.350
Beta blocker, *n* (%)	6 (25)	3 (25)	1
Calcium channel blocker, *n* (%)	7 (29)	2 (17)	0.421
Statin, *n* (%)	2 (8)	1 (8)	1
Hemoglobin, gr/dL, (mean ± SD)	12.7 ± 1.01	13.2 ± 1.3	0.046
Number of diseased vessels, (mean ± SD)	2.71 ± 0.65	2.63 ± 0.93	0.609
Ejection fraction, % (mean ± SD)	48 ± 7	49 ± 8	0.352
Contrast volume, mL (mean ± SD)	55 ± 5.9	61.7 ± 3.9	0.001

ACE-ARB: Angiotensin converting enzyme inhibitor, Angiotensin receptor blocker; CCC: Coronary collateral circulation; SD: Standart deviation; CAN: Contrast associated nephropathy.

**Table 2 medicina-56-00099-t002:** Coronary collateral circulation and Contrast associated nephropathy.

	CAN (−) Group *n* = 24	CAN (+) Group *n* = 12	*p*
Poor CCC, *n* (%)	12 (50)	9 (75)	0.282 *
Good CCC, *n* (%)	12 (50)	3 (25)	

* Fisher’s exact test; CCC: Coronary collateral circulation.

**Table 3 medicina-56-00099-t003:** Logistic regression analysis.

	*p*	Odds Ratio	95%CI
Contrast volume (mL)	0.026	1.4	1.037–1.799
Basal creatinine (mg/dL)	0.038	4.2	1.797–41
CCC	0.472	0.348	0.020–6.176
BMI (kg/m^2^)	0.279	1.5	0.729–2.999
Hypertension	0.383	3.587	63–212

BMI: Body mass index; CCC: Coronary collateral circulation; CI: Confidence interval.

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
