# Peer review of "The Relationship Between Contrast Associated Nephropathy and Coronary Collateral Circulation in very Old Patients"

_medicina, 2020, doi:10.3390/medicina56030099_

Round 1
Reviewer 1 Report
The manuscript submitted for consideration by Yildirim et al., report that there is no correlation between coronary collateral circulation and contrast associated nephropathy. Thought the correlation was not found, the study still contributes to our knowledge about the relation between these two conditions. The study design is fair and efficient. However, the authors need to elaborate the introduction (current knowledge and background) and conclusion. Also, the language of the manuscript needs to be improved to the publishing standards. It's better to get it proof read by a native speaker or use the journal language editing service.
Author Response
Query : The authors need to eloborate the introduction and conclusion.
Response : As suggested by the 1st reviewer, we expanded the introduction and added 3 current literature
Reviewer 2 Report
This article is well-written.
Please use abbreviations of CAN and CCC throughout the paper, as these both terms are already explained in line 33 and 40. For example at line 41, use abbreviation instead of contrast associated nephropathy. Line 45, remove space; ‘impaired vasoconstriction/vasodilation balance….’ Section 2.1: add detailed information about the patients considered for this study. Line 84 and 96: use abbreviation CAN instead of contract associated nephropathy. About the patients considered for the study, categorize the gender also. And also compare the effects based on gender. Line 114: ‘In the CAN (-) group, 12 patients (57%) had poorly developed….., while 12 patients (80%)….’
There might be some mistake in the number of patients, please check.
Author Response
Dear Editor,
Query : The editör didn’t find approval ID and date in our article.
Response: We have pointed out the number and date of the ethical committee in the first line of the patients section of the material method section.
Query : Please usethe abbreviations of CAN and CCC throughout the paper.
Response :We have applied the Contrast Associated Nephropathy as CAN and coronary collateral circulation as CCC throughout the manuscript on the recommendation of the reviewer 2.
Query : Line 45 remove space,’impaired vasoconstriction/vasodilatation balance’
Response : We fixed the error by removing a space as impaired vasoconstriction / vasodilatation balance’
Query : There might be some mistake in the number of patients in line 114.
Response : We corrected the number and percentage of patients in the table and article. This error occurred due to incorrect markup in crostab (row instead of column), but statistical p value is the same.
We described all the corrections in red and reloaded them with word format.
I hope that reviewing process finds the manuscript acceptable for publication in the journal.
Sincerely Yours,
Tarik Yildirim, MD
Department of Cardiology, Balikesir University School of Medicine Balikesir, Turkey kdrtarik@gmail.com
Round 2
Reviewer 1 Report
The authors have addressed previous concerns. No further corrections seem necessary.